# A Structural and Dynamic Analysis of the Partially Disordered Polymerase-Binding Domain in RSV Phosphoprotein

**DOI:** 10.3390/biom11081225

**Published:** 2021-08-17

**Authors:** Christophe Cardone, Claire-Marie Caseau, Benjamin Bardiaux, Aurélien Thureaux, Marie Galloux, Monika Bajorek, Jean-François Eléouët, Marc Litaudon, François Bontems, Christina Sizun

**Affiliations:** 1Institut de Chimie des Substances Naturelles, CNRS, Université Paris Saclay, 91190 Gif-sur-Yvette, France; cardone.christophe@gmail.com (C.C.); cmcaseau@gmail.com (C.-M.C.); marc.litaudon@cnrs.fr (M.L.); francois.bontems@cnrs.fr (F.B.); 2Structural Bioinformatics Unit, Department of Structural Biology and Chemistry, Institut Pasteur, CNRS UMR3528, 78015 Paris, France; bardiaux@pasteur.fr; 3Soleil Synchrotron, SWING, 91190 Saint-Aubin, France; aurelien.thureau@synchrotron-soleil.fr; 4Unité de Virologie et Immunologie Moléculaires, INRAE, Université Paris Saclay, 78352 Jouy-en-Josas, France; marie.galloux@inrae.fr (M.G.); monika.bajorek@inrae.fr (M.B.); jean-francois.eleouet@inrae.fr (J.-F.E.)

**Keywords:** respiratory syncytial virus, phosphoprotein, nuclear magnetic resonance, tetramerization domain, transient secondary structure, protein dynamics

## Abstract

The phosphoprotein P of *Mononegavirales* (*MNV*) is an essential co-factor of the viral RNA polymerase L. Its prime function is to recruit L to the ribonucleocapsid composed of the viral genome encapsidated by the nucleoprotein N. *MNV* phosphoproteins often contain a high degree of disorder. In *Pneumoviridae* phosphoproteins, the only domain with well-defined structure is a small oligomerization domain (P_OD_). We previously characterized the differential disorder in respiratory syncytial virus (RSV) phosphoprotein by NMR. We showed that outside of RSV P_OD_, the intrinsically disordered N-and C-terminal regions displayed a structural and dynamic diversity ranging from random coil to high helical propensity. Here we provide additional insight into the dynamic behavior of P_Cα_, a domain that is C-terminal to P_OD_ and constitutes the RSV L-binding region together with P_OD_. By using small phosphoprotein fragments centered on or adjacent to P_OD_, we obtained a structural picture of the P_OD_–P_Cα_ region in solution, at the single residue level by NMR and at lower resolution by complementary biophysical methods. We probed P_OD_–P_Cα_ inter-domain contacts and showed that small molecules were able to modify the dynamics of P_Cα_. These structural properties are fundamental to the peculiar binding mode of RSV phosphoprotein to L, where each of the four protomers binds to L in a different way.

## 1. Introduction

A number of viral proteins are intrinsically disordered proteins (IDPs) or contain large intrinsically disordered regions (IDRs) [1,2,3]. Compared to globular proteins or protein domains, IDPs and IDRs sample different conformations in their free form and even in bound forms [4,5,6]. They can contain several binding sites, often in the form of short linear motifs of rather weak affinity, but still marked specificity [7,8]. Their flexibility is key for their functional versatility. In the case of viral IDPs, this structural adaptability compensates for the reduced number of proteins encoded by a viral genome, allowing these proteins to carry out a variety of functions at different stages of the viral cycle ranging from host cell infection, evasion of the immune system, viral genome replication to formation of new viral particles.

The RNA-dependent RNA polymerase (RdRp) complex of viruses of the *Mononegavirales* (*MNV*) order, i.e., non-segmented negative strand RNA viruses, has been widely investigated from a functional and a structural perspective [9,10,11,12,13]. The *MNV* RdRp complex works either in transcription or in replication mode. The *apo* form is minimally composed of two viral proteins: the large catalytic subunit (L protein) and its phosphoprotein co-factor (P protein). Its template is a ribonucleoprotein complex, formed by a genomic RNA molecule embedded in a sheath made of viral nucleoprotein (N protein). Recognition between both complexes, which leads to formation of the *holo* RdRp complex, is mediated by a direct interaction between N and P proteins [11,14,15,16]. The P protein not only acts as a linchpin for the *holo* RdRp complex, but also serves as an adaptor protein that tethers additional co-factors to the RdRp complex [17].

Phosphoproteins but also nucleoproteins of several *MNV* viruses, and more particularly viruses of the *Paramyxoviridae* family, were reported to contain large IDRs, either by bioinformatic analysis or by biochemical and biophysical approaches [18,19,20]. For example, the 525-residues N protein of measles virus (MeV) contains a 120 residues-long C-terminal tail, and MeV P protein contains 380 disordered residues out of 507 [21]. IDRs were analyzed in solution by nuclear magnetic resonance (NMR) and/or small angle X-ray scattering (SAXS), which revealed their conformational versatility [22,23] and conformational trajectories during binding events [24]. Notably, it has been shown for several MNVs, that in infected cells, viral RNA synthesis takes place in membraneless cytoplasmic inclusions with liquid-like properties, where N and P proteins are concentrated [25,26]. Formation of these viral factories was postulated to rely on a liquid-liquid phase separation mechanism, induced by weak interactions driven by the IDRs present on N and P proteins as well as RNA [1,25,26,27]. MeV and RSV N and P proteins also trigger liquid-liquid phase separation in vitro, and recent biochemical and biophysical characterization of N/P induced membraneless organelles confirmed the critical role of P and/or N IDRs for this mechanism in vitro and in infected cells [28,29].

Respiratory syncytial virus (RSV) is a *MNV* virus that belongs to the *Pneumoviridae* family and *Orthopneumovirus* genus. RSV is a major pathogen responsible for pneumonia in children [30] and a leading cause of hospitalization and pediatric mortality due to lower respiratory tract infections worldwide [31]. Its genome encodes 11 proteins [32]. Two of these, the phosphoprotein and the glycoprotein, were predicted to contain high disorder [33], with >50% of residues above the disorder score of 0.5 defined by the PONDR algorithm [34]. No high-resolution 3D structure is available for the RSV G protein. The 241-residue RSV P protein was shown to form stable tetramers via a central oligomerization domain (P_OD_, residues 131–150), tetramerization being critical for its functionality [35]. RSV P_OD_ is flanked by large disordered N- and C-terminal domains (P_NT_, residues 1–130, and P_CT_, residues 151–241, Figure 1A) with locally high α-helical propensity [35,36,37,38,39,40].

High resolution insight into the structure of P became available when the structure of the RSV L—P complex was solved by cryo-EM [41,42]. The latter provided a structural basis for the polymerase and mRNA capping catalytic activities of RSV L [41]. It also revealed a “tentacular” binding mode of P to L, where each P protomer binds differently to L [41]. Even though the full-length P was engaged in the cryo-EM L—P complex, only the tetramerization domain and a large C-terminal part (residues 151–183, 151–186, 151–199 and 151–228, depending on the P protomers, Figure 1B) were resolved. Notably P_NT_ was missing in the cryo-EM maps.

We previously analyzed RSV phosphoprotein by NMR to acquire a more detailed structural insight into its disordered regions [17,43]. We showed that these regions displayed very different dynamic behaviors. P_NT_ is nearly fully flexible, except for two regions with low α-helical propensity. In contrast, P_CT_ is structurally heterogeneous. It contains a domain with marked α-helical propensity, which is located immediately C-terminally to P_OD_ (P_Cα_, residues 165–205). P_Cα_ is followed by a fully disordered C-terminal tail (P_Ctail_, residues 210–241). By comparing our NMR results with the cryo-EM structures of the RSV L—P complex, it appears that P_Cα_ coincides with the major part of the L-binding site.

NMR helped rationalize the relationship between the ability of RSV phosphoprotein to recruit multiple protein partners to the RdRp complex, e.g., viral and cellular proteins required for efficient transcription [44], and several conserved small linear motifs with α-helical or β-strand secondary structure propensity in P_NT_ and P_Ctail_ [17,43]. However, the P_Cα_ domain could not be fully characterized by NMR. This is inherent to this technique, as NMR signals are severely broadened and ultimately become undetectable when protein dynamics or exchange phenomena take place on a µs–ms timescale, precluding further analysis. This left open the question how the structure of RSV free P in solution relates to that of P in complex with L, and more particularly if the different conformations observed for the four protomers of RSV P in the L—P complex pre-exist in solution. To answer these questions, we resorted again to NMR, complemented with other biophysical techniques, such as SAXS and Dynamic Light Scattering (DLS). By using small phosphoprotein fragments centered on or adjacent to P_OD_, we found out that NMR signals were recovered for P_Cα_ as well as P_OD_ domains. This allowed us to establish a structural and dynamic picture of these two domains and to acquire new insight into their interactions in solution. We also showed, by analyzing binding of the small molecule garcinol, that the dynamics of P_Cα_ can be interfered with, which opens new avenues for targeting the oligomeric structure of RSV phosphoprotein as a pharmaceutical target.

## 2. Materials and Methods

### 2.1. Expression and Purification of Proteins

RSV phosphoprotein (P) and fragments of P were expressed in *E. coli* BL21(DE3) with an N-terminal GST-tag, using plasmids derived from a pGEX vector, containing ampicillin resistance, as described previously [35]. ^15^N- and ^15^N^13^C-labeled P protein samples were produced in 1 L of minimum M9 medium supplemented with 1 g∙L^−1 15^NH_4_Cl (Eurisotop, Saint-Aubin, France) and 4 g∙L^−1^ unlabeled or 3 g∙L^−1 13^C-labeled glucose (Eurisotop, Saint-Aubin, France). Expression was induced with 80 µg∙L^−1^ IPTG overnight at 28 °C. Bacteria were harvested and disrupted (Constant Systems Ltd., Daventry, UK) in 50 mM Tris pH 7.8, 60 mM NaCl, 0.2% Triton X-100 lysis buffer supplemented with protease inhibitors (complete EDTA-free, Roche, Boulogne-Billancourt, France). After clarification by ultracentrifugation, 2 mL of glutathione Sepharose resin beads (GE Healthcare, Buc, France) were added to the supernatant, and the mixture was incubated overnight at 4 °C. The resin beads were then washed with thrombin cleavage buffer (20 mM Tris pH 8.4, 150 mM NaCl, 2.5 mM CaCl_2_) before addition of 5 units of biotinylated thrombin (Novagen, Merck Millipore, Guyancourt, France). After incubation for 16 h at 4 °C, the cleaved protein was eluted and thrombin was captured with streptavidin resin according to the manufacturer’s instructions (Novagen, Merck Millipore, Guyancourt, France). In a final step, samples were dialyzed into NMR buffer (20 mM sodium phosphate, pH 6.5, 100 mM NaCl) and concentrated to 50–300 µM on Amicon Ultra centrifugal filters with 10 kDa cut-off (Merck Millipore, Guyancourt, France). Some samples were exchanged into D_2_O buffer (prepared by reconstituting lyophilized NMR buffer into 100% D_2_O (Eurisotop, Saint-Aubin, France) by passing 4 times through a desalting column (Biospin, Bio-Rad, Les Ulis, France) equilibrated in this buffer. All samples were analyzed by SDS-PAGE. Protein concentrations were determined from Bradford or BCA assays and by measuring the absorbance at 280 nm for proteins containing tyrosine residues.

### 2.2. Nuclear Magnetic Resonance (NMR) Spectroscopy and Chemical Shift Assignments

2D and 3D NMR experiments for backbone assignment were carried out on Bruker Avance III spectrometers at magnetic fields of 14.1, 16.4 or 18.8 T (600, 700 or 800 MHz ^1^H Larmor frequencies, respectively), equipped with cryogenic TCI probes. Samples contained 7% D_2_O to lock the magnetic field. ^1^H and ^13^C chemical shifts were referenced to 4,4-dimethyl-4-silapentane-1-sulfonic acid (DSS). Spectra were processed with Topspin 4.0 (Bruker Biospin, Wissembourg, France) and analyzed with CCPNMR 2.5 [45] software.

Sequential backbone assignment of ^15^N^13^C labeled proteins was carried out using BEST-TROSY versions of triple resonance 3D HNCO, HNCA, HN(CO)CA, HNCACB, HN(CO)CACB experiments [46] and standard versions of 3D CBCA(CO)NH and HBHA(CO)NH experiments. Amide ^1^H/^15^N and ^1^H_α_ assignments of ^15^N labeled proteins was carried out by recording 3D ^15^N-separated DIPSI-HSQC and NOESY-HSQC spectra. A 3D (H)CCH-TOCSY spectrum was acquired at 800 MHz on a P_127–163_ sample in 100% D_2_O buffer for sidechain assignment.

### 2.3. High-Pressure NMR Measurements

Steady-state high pressure NMR measurements were carried out in a zirconia ceramic NMR tube (Daedalus Innovations, Aston, PA, USA) filled with 350 µL of 400 µM ^15^N^13^C labeled P_127–205_ and 20 µL D_2_O. Pressure was generated with an X-treme 60 syringe pump (Daedalus Innovations) filled with mineral oil. The temperature was set to 313 K. ^1^H-^15^N HSQC spectra were acquired at variable pressure between 1 bar and 2.5 kbar, with 100–500 bar increments. 3D HNCA spectra used for chemical shift assignment were acquired at 500 and 2500 bar.

### 2.4. NMR Structure Determination of P_127–163_

Distance restraints to calculate the NMR structure of P_127–163_ were derived from NOESY experiments performed on two samples: a uniformly ^15^N^13^C-labeled sample and a sample with mixed labeling. The latter was reconstituted from an equimolar mixture of unlabeled and ^15^N^13^C-labeled P_127–163_ incubated overnight at 4 °C with 6 M guanidinium chloride. The sample was then dialyzed at 4 °C for two hours on a membrane with a 3.5 kDa cut-off, successively into NMR buffer supplemented with 8 M urea (200 mL), NMR buffer with 2 M urea (3 × 500 mL) and NMR buffer (3 × 1 L).

Intra- and inter-protomer distance restraints were obtained by measuring the signal heights in two 3D NOESY-HSQC experiments with 80 ms mixing time acquired on the P_127–163_ sample with uniform labeling: a ^15^N-separated NOESY-HSQC in H_2_O buffer and a ^13^C-separated NOESY-HSQC in D_2_O buffer. They were treated as ambiguous restraints. Unambiguous inter-protomer distances were obtained from a ^13^C-separated NOESY-HSQC with a ^13^C-filter in the t1 dimension, measured on the mixed sample in D_2_O buffer. All NOESY data were acquired at a magnetic field of 22.3 T (950 MHz ^1^H Larmor frequency). Dihedral angle restraints were obtained from backbone chemical shifts by using the TalosN software [47]. 3D structure calculation of tetrameric P_127–163_ was performed with the ARIA 2.3.2 software [48] imposing a C4 symmetry [49].

NMR restraints and structural statistics for RSV P_127–163_ were generated with ARIA and the Protein Structure validation suite V1.5 (http://psvs.nesg.org/, http://rtti7.uv.es/, accessed on 15 April 2020).

### 2.5. Paramagnetic Spin Labelling and Paramagnetic Relaxation Enhancement (PRE) Measurements

Two serine to cysteine mutants of P_127–205_, S143C and S156C, were constructed using the QuickChange mutagenesis kit (Stratagene, Agilent, Les Ulis, France). The mutations were verified by sequencing. Mutant proteins were expressed and purified such as wild type proteins. After purification, protein samples were treated with 10 mM DTT at room temperature for 2 h to ensure that cysteine residues were in a reduced state. DTT was then removed by passing twice through a Biospin desalting column (Biospin, Biorad, Les Ulis, France) equilibrated with 50 mM Tris, pH 8.0, 200 mM NaCl. The cysteine mutants (200 µL at 300 µM final concentration) were reacted overnight at 15 °C in the dark with 10 molar equivalents of 3-(2-iodoacetamido)-PROXYL radical (IAP, Sigma-Aldrich, Saint-Quentin-Fallavier, France) taken from a 45 mM stock solution in ethanol. Unreacted IAP was removed by passing the samples three times through a desalting column equilibrated in the final NMR buffer. After data acquisition with the paramagnetic IAP spin label, the sample was treated with 10 molar equivalents of ascorbic acid (Sigma-Aldrich, Saint-Quentin-Fallavier, France) at pH 6.5 overnight in the dark to generate a diamagnetic spin label.

Paramagnetic relaxation enhancements (PREs) were determined for each residue as the ratio of peak intensities measured in the ^1^H-^15^N HSQC spectra of samples with spin labels in the paramagnetic versus diamagnetic state (I_para_/I_dia_). NMR data were acquired at a magnetic field of 14.1 T and a temperature of 313 K. To model the PRE profiles, 1000 protein conformers were simulated with the Flexible-meccano software [50], assuming 100% α-helical propensity for residues 130–156 and using the α-helical propensities determined from backbone chemical shifts with TalosN [47] for residues 157–205. The segments 143–156 (containing the labeled positions) and 160–199 (where strong PREs were measured) were defined as contact regions. PREs were simulated with no contact and contact distances of 7, 10 and 15 Å.

### 2.6. Small Angle X-ray Scattering (SAXS)

SAXS experiments were performed at the SWING beamline at the SOLEIL synchrotron (Saint-Aubin, France), using an EigerX 4M detector at a distance of 2 m. The temperature was set to 293 K. P_127–163_ and P_127–205_ solutions were prepared in 20 mM Na phosphate pH 6.5, 100 mM NaCl buffer at final concentrations of 200 µM and 325 µM, respectively. Protein samples (50 µL) were injected onto a size exclusion column (BioSec3-300, Agilent, Buc, France) and eluted directly into the SAXS flow-through capillary cell at a flow rate of 0.3 mL/min, as previously described [51]. 510 SAXS frames (exposure time: 990 ms, dead time: 10 ms) were collected continuously during elution. Data reduction to absolute units, buffer subtraction and averaging of identical frames corresponding to the elution peak (10 and 13 frames for P_127–163_ and P_127–205_, respectively) were conducted with the SWING in-house software FOXTROT [52] and US-SOMO [53]. I(0) and Rg values were determined with the PRIMUS software [54]. Normalized Kratky curves were obtained by plotting (q*Rg)^2^*I(q)/I(0) as a function of q*Rg, where q is the scattering vector amplitude. The theoretical normalized Kratky plot for a globular protein was calculated using the expression (q*Rg)^2^*exp(-(q*Rg)^2^/3). The SAXS data intensity scattering curve of P_127–163_ was fitted on the FoXs server [55] using the NMR structure of P_127–163_ and declaring the 9 N- and 15 C-terminal residues as flexible.

### 2.7. Interaction Experiments with Garcinol

Stock solutions of garcinol (CAS number 78824-30-3) were prepared in DMSO (3 mM) or in ethanol (30 mM) and added to 150 µM P_161–241_ and 300 µM P_127–205_, up to a molar ratio of 3:1 and 1:1, respectively. The mixtures were analyzed by NMR at a Larmor frequency of 600 MHz ^1^H and at a temperature of 293 K for P_161–241_, and at 700 MHz and 313 K for P_127–205_.

### 2.8. Dynamic Light Scattering (DLS)

Dynamic light scattering measurements were carried out on a Zetasizer Nano S instrument (Malvern, Orsay, France) at a backscatter angle of 173° and a temperature of 298 K. P_127–205_ samples, used for NMR before, were centrifuged before being transferred into disposable 40 µL cuvettes (Brand Z637092, Wertheim, Germany). Size distribution analysis was performed within the Zetasizer software, by setting the refractive index to 1.450 and the viscosity to 0.8872 cP.

### 2.9. Illustrations

Figures were prepared using Pymol [56].

### 2.10. Data Depositions

RSV P_127–163_ chemical shifts were deposited at the BMRB with accession number 34513. Coordinates and restraints for the NMR structure of RSV P_127–163_ were deposited at the Protein Data Bank under accession code 6YP5. SAXS data for P_127–163_ were deposited at the SASBDB with accession code SASDJD2.

## 3. Results

### 3.1. Signal Broadening in NMR Spectra of RSV Phosphoprotein Fragments Reveals Regions with Partial to High Structural Order

2D ^1^H-^15^N HSQC spectra provide structural and dynamic fingerprints of ^15^N-labeled proteins at the single residue level, as each amide pair accounts for an individual signal, identified by its ^1^H and ^15^N chemical shifts. As the chemical shift of an atomic nucleus is sensitive to its electronic environment, it is a highly sensitive local probe for structural complexity. As IDPs and IDRs lack defined 2D or 3D structure, they display a narrow amide ^1^H chemical shift dispersion. Moreover, IDP/IDR signals are sharp due to rapid motions of the polypeptide chain. In contrast, the amide ^1^H chemical shift dispersion of well-structured protein regions is wider. Signal linewidths depend on nuclear relaxation, which depends on molecular tumbling. Hence, as a general rule, linewidths increase concomitantly with molecular size, but conformational and chemical exchange contribute to additional line broadening.

We previously measured ^1^H-^15^N HSQC spectra for the RSV full length P. One third of the signals was missing due to severe line broadening [43]. We identified the missing signals as those of the oligomerization domain P_OD_ and of the P_Cα_ domain. When we measured ^1^H-^15^N HSQC spectra of RSV P N- and C-terminal fragments containing P_OD_, the same signals were missing [43]. P_Cα_ signals were only detected in monomeric fragments without P_OD_. This raised the question if the structural and dynamic information obtained from these monomeric fragments by chemical shift and nuclear relaxation analysis was relevant for the tetrameric protein.

We therefore sought new conditions to recover P_OD_ and P_Cα_ signals for tetrameric fragments in amide-detected NMR experiments. We produced eight ^15^N-labeled fragments of RSV phosphoprotein of various sizes (Figure 1A). The P_127–163_ fragment roughly corresponds to P_OD_. The P_113–163_ fragment has an N-terminal extension with high sequence conservation within the *Orthopneumovirus* genus. P_127–184_ has a C-terminal extension that is conserved throughout the *Pneumoviridae* family, whereas the longer P_127–205_ fragment is only well conserved within *Orthopneumoviruses* [17]. P_127–229_ contains the full L-binding site revealed in the cryo-EM RSV L—P complex structure [41,42]. P_127–241_, which consists of P_OD_ and the entire C-terminal domain, was investigated previously and reproduced the behavior of full-length RSV phosphoprotein [43]. In addition, we produced two C-terminal fragments without P_OD_, P_161–205_ and P_161–229_, for comparison.

We measured their ^1^H-^15^N HSQC spectra under identical buffer and temperature conditions. We first recorded NMR data at low temperature, i.e., at 288 K (Figure 2A). This condition is more favorable to observe amide signals that are not or only loosely involved in hydrogen bonds, i.e., signals from IDRs. The signals were sharp and not well dispersed, which pointed at a disordered state of the protein fragments. For most fragments, the number of observed signals was less than expected. For P_113–163_, only a set of ~20 sharp signals was observed, corresponding to residues 113–130 and 158–163. The signals of residues 131–157 were 5 to 10-fold less intense than the set of sharp signals; they became visible by lowering the contour levels close to the noise level. For P_127–184_, ~25 sharp signals assigned to residues 157–184 were observed, but the signals from P_OD_ were missing. For P_127–205_ only a few flexible residues of the N-and C-terminal extremities were visible. For P_127–229_ and P_127–241_, ~30 and ~40 resonances were detected; they were assigned to residues 204–229 and 203–241, respectively, and correspond to P_Ctail_. In contrast, no signal was missing for the two constructs without P_OD_, P_161–229_ and P_161–205_. The results are in overall agreement with previous results obtained with longer fragments including P_1–127,_ P_1–163_ and P_161–241_ [43]. Comparison between P_127–229_ and P_161–229_ (superimposed in Figure 2A) indicates that the chemical shifts of P_Ctail_ are independent of the presence of P_OD_, confirming that this part of RSV P is a fully disordered IDR.

In contrast to P_Ctail_ signals, P_Cα_ signals were broadened out in nearly all P_OD_–containing fragments, with the notable exception of P_127–184_. In the latter case, the sharpness of P_Cα_ signals indicates that residues 157–184 were disordered and flexible, as a consequence of truncation within the P_Cα_ region. Hence, P_127–184_ is not representative of the full-length protein, contrary to P_127–205_.

The signals of P_OD_ were also broadened out, except in P_127–163_, a P_OD_–containing fragment with very short N- or C-terminal extensions. The amide ^1^H chemical shifts of P_127–163_ were only little more dispersed than those of fully disordered regions, but this remains compatible with helical secondary structure. Linewidths in P_127–163_ were broader than for RSV P IDRs, which is indicative of the presence of a stably folded domain. Together these results suggest that extensions longer than 10 residues induce motions on the µs–ms time scale, presumably shearing motions in the coiled coil P_OD_, resulting in signal broadening.

In addition to NMR data at a temperature of 288 K, we acquired data at 313 K to take advantage of signal narrowing for folded protein signals due to less efficient nuclear relaxation at higher temperature. We focused on the three P_OD_–containing fragments with C-terminal extensions: P_127–163_, P_127–184_ and P_127–205_ (Figure 2B). At 313K, the P_OD_ signals (residues 130–154) were indeed observable and superimposed well between the three fragments, indicating that P_OD_ adopted the same structure. The P_Cα_ signals of P_127–184_ and P_127–205_ were also observed, but did not superimpose, confirming that the residues 160–184 adopt different conformations in both fragments. P_OD_ signals were less intense and broader than those of the C-terminal extensions in the P_127–184_ and P_127–205_ fragments, which indicates that P_Cα_ is not stably associated, neither in a coiled coil conformation such as P_OD_, nor directly bound to P_OD_.

### 3.2. Secondary Structure of RSV P_OD_ and P_Cα_ by NMR

To gain more insight into the structures of P_127–184_ and P_127–205_, we analyzed secondary structure propensities for each residue based on backbone chemical shifts at 313 K. Chemical shifts were determined by sequential assignment from 3D triple resonance and ^15^N-separated NOESY-HSQC experiments. We used the TalosN server, which predicts α-helix, β-strand and random coil propensities [47]. The input consisted of ^13^Cα, ^13^Cβ, ^13^CO and ^1^Hα chemical shift values. We determined nearly 100% α-helical propensity for the P_OD_ part in both fragments and significant α-helical propensity in the P_Cα_ part (Figure 3A). The short linker region between P_OD_ and P_Cα_, around residue T160, has nearly 100% random coil propensity.

P_Cα_ residues displayed less α-helical (<50%) and more random coil propensity in P_127–184_ than in P_127–205_ at 313 K. This suggests that the 185–205 region promotes upstream α-helix formation in the P_Cα_ domain. The increased disorder of the truncated P_Cα_ domain in P_127–184_ was also found at lower temperature, since at 288 K secondary structure propensities derived from ^13^Cα chemical shifts were mostly random coil for residues 155–184, with <20% α-helical propensity (Figure 3A).

Even though the NMR data of P_127–205_ confirm that P_Cα_ has high α-helical propensity, they also show that P_Cα_ does not form a long stable helix such as P_OD_. In P_127–205_, P_Cα_ consists of two or three regions with α-helical propensities >50%, but still <100% (Figure 3A). Two regions, termed α_C1_ (residues 174–185) and α_C2_ (residues 189–198), coincide with two transient C-terminal α-helices previously determined from the monomeric fragments P_161–241_ and P_1–121 + 161–241_ by solution NMR at 288 K [43].

In a ^15^N NOESY-HSQC experiment performed with P_127–205_, the amide protons of P_OD_ displayed (i, i+1) Nuclear Overhauser Effects (NOEs) indicative of α-helices, but no NOEs with water, showing that they were protected against water exchange and engaged in stably formed α-helices. The P_Cα_ region lacked NOEs indicative of stable helix formation. This comforts the hypothesis that P_OD_ and P_Cα_ form independent structural entities, with different tumbling properties and that the α-helices in P_Cα_ remain transient and dissociated.

It is noteworthy that α_C1_ corresponds to a region that adopts an α-helical conformation in all four P protomers in the RSV L—P complex, whereas α_C2_ is stabilized into α-helix for only two protomers [41,42]. The 163–173 region, which displays less but still significant α-helical propensity in P_127–205_, is stabilized into an α-helix in only one protomer in the RSV L—P complex. In this complex, one P protomer forms two additional C-terminal helices spanning residues 205–211 and 215–233. These do not display any significant α-helical propensity in the free P [43], suggesting that they only fold on binding. Our results thus show that the different conformations observed for the four protomers of RSV P in the L—P complex partly pre-exist in solution. These helices are on the most ordered side of the spectrum of IDRs/IDPs, but remain transient in solution.

Furthermore, we found that residues 135–154, located in the P_OD_ region of P_127–205_, have ~100% α-helical propensity. The coiled coil core domain in solution thus appears to be slightly shifted with respect to the tetrameric core of P in the cryo-EM L—P structure. The latter comprises only residues 131–150, but three protomers remain associated up to residue S156, which is closer to the boundaries found in solution. To test the hypothesis of helix fraying in P_OD_, we performed amide ^1^H/^2^H exchange with the P_127–163_ fragment. ^1^H/^2^H exchange results in vanishing ^1^H-^15^N HSQC signals for amides that are not involved in strong hydrogen bonds. As expected, the N- and C-terminal ends of P_127–163_, spanning residues D129–R134 and V154–R163, respectively, were fully ^1^H/^2^H exchanged (Figure 3B). Only a few amides located in the middle of the sequence (D136, D139–L142, I145–L146 and L149) were completely exchange-protected. L135, I138, M148, L152 and V153 amides were partially protected, which suggests that the helices start fraying at both ends. Surprisingly, several other central residues (R137, S143–E144, G147, H150 and T151) were nearly completely exchanged, suggesting that the P_OD_ helices are not perfectly straight and contain kinks that account for solvent accessibility in the 137–151 region.

### 3.3. Probing the Stability of the RSV P_OD_–P_Cα_ Region by High Pressure NMR

High pressure NMR allows to locally probe protein unfolding due to mechanical compression [57]. A general rule is that with increasing pressure, protein conformational equilibria are shifted towards states with smaller partial volume and increasing conformational disorder [58]. Even if high pressure NMR best applies to globular proteins with internal cavities, leading to large partial volume changes, we nevertheless tested this technique on the RSV fragment P_127–205_, i.e., the P_OD_–P_Cα_ region, which already contains a certain amount of disorder. This approach was driven by two considerations. First, solvation plays a role in the pressure response of proteins [58,59], which applies to any kind of protein. Second, protein chemical shifts have been reported to be highly sensitive to pressure [58]. Amide ^1^H and ^15^N chemical shifts are expected to display a linear pressure dependence for rigid proteins and to deviate from linearity for proteins in multiple conformational states [60].

We measured ^1^H-^15^N HSQC spectra of P_127–205_ at a temperature of 313 K, where all signals were observable, and increased pressure from 1 to 2500 bar in 100–500 bar steps. We observed amide chemical shift perturbations (Figure 4A). To correctly re-assign amide chemical shifts at each pressure step, we checked the sequential assignment in two 3D HNCA experiments acquired at 500 and 2500 bar. In addition to chemical shift perturbations, we also observed intensity perturbations. For several residues, e.g., R137 and G147 located in P_OD_, the intensity decreased; for other residues, e.g., T160 and G172 located in P_Cα_, the intensity increased (Figure 4A). This was a first hint that P_OD_ and P_Cα_ could be discriminated by high pressure NMR. These intensity variations suggest that conformational equilibria take place and that they shift with variable pressure. In the case of P_OD_, the coiled coil conformation may be destabilized. In the case of P_Cα_, which contains transient helices, the conformational equilibria that exist at ambient pressure are also shifted, likely towards a more disordered state. However, ^13^Cα chemical shifts measured at 500 and 2500 bar did not show significant perturbations as compared to ambient pressure, indicating that increasing pressure did not significantly affect secondary structure propensities in P_Cα_.

We analyzed amide ^1^H and ^15^N chemical shifts into more details by fitting the pressure dependency with a second-order polynomial. We extracted linear (B_1_) and quadratic (B_2_) pressure coefficients (Figure 4B) and compared them to reference values tabulated for each amino acid type in random coil conformation [60]. ^1^H B_1_ values were slightly higher for P_Cα_ than for P_OD_, but they were all close to the reference values. ^1^H B_2_ values deviated more from the reference values, and they were more negative for P_Cα_ than for P_OD_. For residues in the vicinity of H150, the deviations observed for ^1^H B_2_ may be accounted for by pressure induced pH variations [61]. ^1^H pressure coefficients thus do not allow unambiguously distinguishing between P_OD_ and P_Cα_. ^15^N B_1_ and B_2_ values draw a clearer picture of the structural difference between these domains. For P_OD_, ^15^N B_1_ and B_2_ values were small and close to the reference values. In the P_Cα_ domain, both B_1_ and B_2_ significantly deviated from the reference values. The transient helices α_C1_ and α_C2_ displayed the largest deviations (Figure 4B). Taken together, our results suggest that ^15^N B_1_ and B_2_ pressure coefficients allow discriminating between the isolated P_Cα_ helices and the coiled coil P_OD_, and that transient helices are characterized by high ^15^N B_1_ and B_2_ values.

### 3.4. NMR Structure of P_OD_ and Spatial Excursions of P_Cα_

To acquire further insight into the solution structure of P_OD_, we solved the structure of the P_127–163_ fragment by NMR, using dihedral angle and Nuclear Overhauser Effect (NOE) distance restraints. We assigned >97% of sidechain ^1^H chemical shifts of P_127–163_. The main difficulty was to distinguish between intra- and inter-protomer NOEs due to the oligomeric nature of P_OD_. To overcome this difficulty, we produced three P_127–163_ samples: a first fully ^13^C^15^N labeled sample, a second unlabeled sample and a third mixed unlabeled and ^13^C^15^N labeled sample. To produce the latter, we denatured a 50:50 mixture of unlabeled and ^13^C^15^N labeled P_127–163_ with 6 M guanidinium chloride. After renaturation in a buffer without chaotropic agent, P_127–163_ reassembled with the same protomer arrangement, as judged from ^1^H and ^13^C chemical shifts that remained the same after the denaturation/renaturation treatment. We measured distance restraints for a single protomer from conventional NOESY experiments using the fully ^13^C^15^N labeled P_127–163_ sample. We measured and assigned inter-protomer restraints from an edited/filtered NOESY experiment using the mixed labeled P_127–163_ sample. Of note, the detected NOEs were compatible with a parallel arrangement of the tetramer.

As P_127–163_ displayed a single set of chemical shifts, and since P was proposed to form tetramers [35], we assumed a C4 symmetry for structure calculations with the ARIA software [48], using this symmetry (Figure 5A). The structural statistics are given in Appendix A. The α-helical core domain spans residues N128–A157. The six most C-terminal residues are disordered. An 8-amino acid long N-terminal stretch, which is a leftover from a glycine and serine linker between the GST-tag and the protein sequence after thrombin cleavage, is also disordered. Superimposition of the NMR structure with the cryo-EM structure of P_OD_ in Figure 5B shows that the arrangement of the coiled coil domain is comparable to that in the RSV P—L complex [41,42]. The helices in the NMR structure appear to be more bent, with a small kink at position S143, which reminds of the increased water accessibility observed at this position. The helix length of the NMR structure corresponds to that of the longest helix of the coiled coil domain in the cryo-EM structure (Figure 5B). This suggests that the conformation of the C-terminal end of each bound protomer in the P—L complex is determined by its interactions with L. We calculated the electrostatic surface potential of P_127–163_ using the Delphi software [62] (Figure 5A). The surface of the coiled coil domain at the N-terminal side is charged, both negatively and positively, up to residue E144. The C-terminal side, starting at residue I145, is completely hydrophobic. This confers different interaction properties, notably within the RSV P—L complex. In the cryo-EM P—L complex structure, the N-terminal side of P_OD_ sticks out, whereas the C-terminal side is associated to L.

To confirm the oligomerization state of P_127–163_ in solution we performed SAXS measurements. A Guinier plot afforded a radius of gyration (R_g_) of 21.2 Å, which is roughly in agreement with a coiled coil domain length of ~50 Å measured directly from the NMR structure. We fitted the scattering curve with the structure of P_127–163_ using the FoXs server [55]. Even though the fit was not perfect, a tetrameric structure best fitted the curve, as compared to monomeric, dimeric or trimeric structures (Figure 5C).

Normalized Kratky plots were calculated using the R_g_ and I(0) values obtained with the PRIMUS software [54]: 21.2 Å and 0.0447 for P_127–163_; 36.8 Å and 0.0749 for P_127–205_. The bell-shape of the normalized Kratky plot for P_127–163_ for q.Rg < 6 and the position of the bell maximum at q.Rg ≈ √3 [63] indicate that P_127–163_ is mostly globular (Figure 5C). The deviations from the theoretical curve for a globular protein may be ascribed to the disordered N- and C-terminal residues. The deviation from the bell-shape is larger for the P_127–205_ fragment (Figure 5C). Our SAXS data thus confirm that the P_OD_–P_Cα_ region is not globular and comfort the hypothesis that the P_Cα_ helices are not stably associated in solution.

### 3.5. Paramagnetic Relaxation Enhancement Reports on Transient Contacts between the P_Cα_ and P_OD_ Domains

To further investigate the relative flexibility of the P_Cα_ domain with respect to P_OD_, we measured paramagnetic relaxation enhancements by NMR [64]. We previously applied this technique to full-length phosphoprotein to detect transient contacts within the tetrameric protein. We found transient contacts within the disordered N-terminal domain of the phosphoprotein, but also between P_NT_ and P_OD_ [43]. Here we investigated the P_127–205_ fragment, where the P_OD_–P_Cα_ region can be observed, in contrast to the full-length phosphoprotein.

We produced two P_127–205_ samples with a paramagnetic nitroxide spin label (3-(2-iodoacetamido)-PROXYL radical, IAP) at positions 143 and 156, after mutating serine residues 143 and 156 into cysteines. S143 is located inside the coiled coil domain and its sidechain points outward. S156 is located at the C-terminal end of the coiled coil and oriented inward. We first assessed the influence of the mutations on the structure by measuring ^1^H-^15^N HSQC spectra. The spectra of both S143C and S156C mutants displayed chemical shift perturbations up to six residues from the mutated site (Figure 6A). Otherwise, the spectra overlaid well with the WT spectrum, indicating that the mutations did not disrupt the overall structure of P_127–205_. This was confirmed after treatment of the IAP labeled samples with ascorbate, as the two spectra with diamagnetic spin labels were similar, except for the mutated sites and adjacent regions (Figure 6B). The signals of the diamagnetic spin labeled samples were however broader that those of the label-free samples. Despite overall structure conservation, this suggests that the spin labels induced some structural disorder.

The ^1^H-^15^N HSQC spectra measured for the P_127–205_ S143C and S156C mutants with a paramagnetic spin label displayed severe line broadening (Figure 6B) due the contribution of the paramagnetic center to nuclear relaxation of nuclei within a 15 Å distance [65]. We measured the paramagnetic relaxation enhancement (PRE) for each residue, using the intensity ratio I_para_/I_dia_ of NMR signals measured from paramagnetic and diamagnetic samples (Figure 6C). As expected, signals disappeared for residues up to 15 residues N- and C-terminally to the paramagnetic label position. For S143C, additional severe line broadening was observed in the 165–200 region. This points at transient long range interactions between the P_Cα_ helices and the N-terminal half of P_OD_. For S156C, additional line broadening was observed in the 175–200 region, which is only possible if the P_Cα_ region is highly flexible.

We calculated theoretical PREs with the Flexible-meccano software [50] from an ensemble of 1000 P_127–205_ structures (Figure 6C). The structural models were generated by using the α-helical propensities determined above and a 7 Å contact distance between P_OD_ residues 143–156 and P_Cα_ residues 160–199 (Figure 6C). The calculated PREs approximated rather well the trend observed in experimental data. The dimensions measured for these P_127–205_ structures are ~70 Å, which is compatible with a radius of gyration of 36.8 Å determined by SAXS. The structural ensemble of the P_Cα_ region is diverse with helices of various lengths at different positions. These transient and flexible helices explore a large conformational space with respect to the P_OD_ domain and make transient contacts with P_OD_, which lead to overall compaction of P_127–205_.

### 3.6. Effect of Small Molecule Binding on the Dynamic Behavior of RSV P

Garcinol is a polycyclic polyprenylated acylphloroglucinol (PPAP) (Figure 7A). This natural product was isolated from *Moronobea coccinea* (Clusiaceae), a plant of French Guiana [66]. The plant is used in traditional medicine for its antioxidant and potentially antitumor activities [67]. Garcinol was reported to display activity against infectious agents such as Influenza A virus [68]. We identified garcinol as a potential RSV N-P interaction inhibitor by in vitro screening of an in-house chemical library. Garcinol exhibited cytotoxicity, but the in vitro results raised the question if it could directly interact with RSV N or P proteins.

To probe the interaction with RSV P, we first used the P_161–241_ fragment that contains an N-binding site relevant for RSV *holo* polymerase complex formation [69]. We measured an ^1^H-^15^N HSQC spectrum after addition of 3 molar equivalents of garcinol in DMSO and compared it to the reference spectrum of P_161–241_ with an equivalent volume of DMSO (Figure 7B). The temperature was set to 293 K, since we showed previously that this fragment consists of transient helices and disordered regions [43]. The signals of the fully disordered C-terminal tail remained unaffected (Figure 7B,C). In contrast, the whole set of signals corresponding to the P_Cα_ domain (40 residues) displayed severe line broadening and reduced intensities (Figure 7B,C). This was surprising, since the size of the perturbed area is much larger than expected for the binding site of a 600 Da molecule.

We wondered if garcinol would have the same effect on a P_OD_-containing fragment. To answer this question, we used the P_127–205_ fragment that lacks the P_Ctail_, which remained unaffected by garcinol in the P_161–241_ construct. We worked under experimental conditions where the P_OD_ signals were detectable, i.e. at a temperature of 313 K. Garcinol also induced severe line broadening at a 1:1 molar ratio in the whole P_Cα_ domain (Figure 7B,C), such as in the P_161–241_ fragment.

Line broadening may arise from conformational exchange on the µs–ms timescale. In the case of P_Cα_, which already contains transient helices, this would imply that garcinol triggers a change in local dynamics, either by affecting the exchange rate or by shifting the conformational equilibrium towards more order or disorder. Line broadening may also result from a change in molecular tumbling associated with formation of bigger sized particles. Garcinol induced line broadening in P_Cα_ reminds of that observed for the P_OD_ region, when it is flanked by large N- or C-terminal regions. This would imply that garcinol stabilizes the P_Cα_ helices and/or promotes interprotomer or intermolecular interactions between these helices, leading to the observed effect. Our experiments did not permit the determination of the binding site of garcinol nor the stoichiometry of the putative garcinol-P_Cα_ complex.

To acquire insight into potential self-association, we performed DLS measurements with P_127–205_ after NMR measurements. From the size distribution by volume analysis, we deduced a hydrodynamic radius of ~32.5 Å (6.5 nm size, Figure 7D), which is in accordance with the 36.8 Å radius of gyration determined by SAXS. In the presence of ethanol and garcinol/ethanol, the apparent size (equivalent to a diameter) increased from 65 Å to 75 Å and 90 Å, respectively (Figure 7D). In the presence of garcinol/ethanol, a ~100 nm sized species was also observed in the distribution by intensity. However, according to the distribution by volume, it remains a minor species. We expect that an inter-protomer association would lead to compaction rather than to expansion of the P_127–205_ structure. Destabilization of the helical structure could increase the hydrodynamic radius. However, an order-to-disorder transition would not lead to the nearly complete line broadening observed for P_161–241_. The DLS data thus rather suggest that garcinol induces formation of higher order oligomeric forms of P_127–205_, which are in equilibrium with the tetrameric form. We made a last control by adding the RSV N protein in the form of N-RNA rings to the P_161–241_ fragment in the presence of garcinol. N bound to P exactly in the same manner as without garcinol, by involving the last C-terminal residues of P [43].

Taken together these results indicate that the dynamic state and the apparent size of P_127–205_ are modulated by the presence of a molecule such as garcinol. From our data we cannot conclude on the exact molecular mechanism. However, changes are mediated by the isolated transient helices in the P_Cα_ domain. Given the importance of P_Cα_ for the formation of the RSV P-L complex, such molecules may be used to interfere with the viral polymerase by precluding the P-L interaction.

## 4. Discussion and Conclusions

The RSV phosphoprotein acts as a hub protein for the viral RNA dependent RNA polymerase complex by assembling its different components [17]. It binds to the catalytic subunit L on the one hand and to the nucleoprotein enchasing the viral RNA genome on the other hand to form the holo polymerase complex [70,71]. It recruits additional factors such as the RSV M2-1 transcription antiterminator to the polymerase complex [72]. It serves as a docking platform for tertiary complexes such as the complex formed with the protein phosphatase 1 and M2-1, leading to dephosphorylation of M2-1 [44]. The phosphoprotein also plays a role in viral budding. We have recently shown that P directly interacts with the RSV matrix protein and that the tetrameric nature of RSV P is a requirement for this function [73].

The RSV P protein has long eluded structural elucidation due to its intrinsic disorder [36,39,74]. We and others investigated its overall structural and dynamic properties in solution as well as disorder-order transitions of its different domains using several biophysical approaches that reported either on domain properties or individual amino acids in the case of NMR [38,43]. Recently, high-resolution structural data was obtained by cryo-EM for the RSV polymerase, including RSV P, as the main co-factor of the large catalytic subunit L [41,42]. The L—P complex structure highlighted two structural aspects about the P protein. First, its N-terminal domain was invisible in the cryo-EM maps, indicating that it remained disordered and unbound. Second, the domain with high α-helical propensity, P_Cα_, became ordered upon binding L, but each protomer adopted a specific binding mode, with different structures, i.e., containing a different number of α-helices with different boundaries and different binding sites (Figure 1B) [17,41].

Here we investigated into more details the structure and dynamics of the consecutive P_OD_ and P_Cα_ domains in unbound RSV P. P_Cα_ is a neither fully disordered nor globular region, but situated in the continuum between these two extremes (Figure 3 and Figure 4). P_OD_ is the only stably folded part of the protein. However, even the boundaries of P_OD_ are fluctuating. Structure resolution by solution NMR showed that the C-terminal end of the coiled coil tetramer may extend up to residue V154 (Figure 5), whereas in the L—P complex, one protomer already disengages from the tetramer one helix turn before. We showed that transient α-helices were formed in unbound P protein, by using tetrameric fragments of P. In previous studies we had shown that these helices also exist in monomeric fragments [43]. We moreover showed that the full P_Cα_ domain was necessary to promote α-helicity, since P_Cα_ did not adopt its native conformation in the truncated P_127-184_ fragment (Figure 3). The boundaries of these transient helices are delineating those of the stabilized helices in the L—P complex. Interestingly the α_C1_ region displayed the highest α-helical propensities (Figure 3). α_C1_ was also singled out in high-pressure NMR experiments by higher linear and quadratic pressure coefficients (Figure 4). The α_C1_ region is the only one adopting α-helical secondary structure in all four protomers in the RSV L—P complex [41,42]. From our data, we conclude that the bound conformation of α_C1_ pre-exists in unbound P, in equilibrium with a more disordered state shifted towards the ordered state.

By using paramagnetic labels, we showed that the P_Cα_ domain makes transient contacts with P_OD_, leading to compaction of the P_127–205_ fragment. The overall size of the structural models derived from PRE analysis (Figure 6) agrees with the radius of gyration determined by SAXS and the size measured by DLS. Since we previously showed that the last 30–35 C-terminal residues of the P protein, corresponding to P_Ctail_, are fully disordered [43], we may infer that these transient contacts and overall compaction, which are inherent to the tetrameric nature of RSV P, also take place in the full-length protein.

Finally, we showed that the P_Cα_ domain might be targeted by small molecules to modify its dynamics Figure 7), by promoting self-association of the free helices from different protomers or by promoting oligomerization of P. The structural plasticity of the P protein, and in particular of its P_Cα_ domain, plays a crucial role for the L—P complex formation, i.e., for a functional viral polymerase. This now needs to be verified for the full-length P protein using molecules with demonstrated antiviral activity.

## Figures and Tables

**Figure 1 biomolecules-11-01225-f001:**
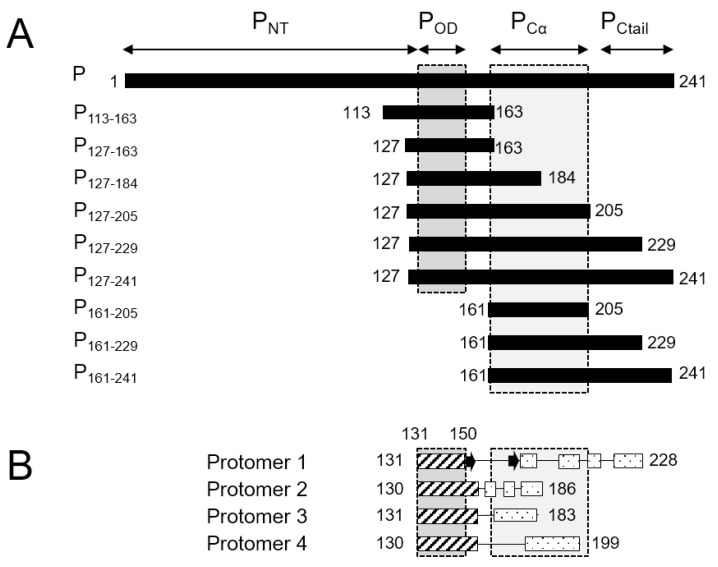
Schematic representation of RSV phosphoprotein (P) domains and fragments. (**A**) Boundaries of the N-terminal domain, oligomerization domain, C-terminal α-helical domain and C-terminal tail of the RSV P protein and definitions of the RSV fragments used herein. (**B**) Topology (α-helices and β-strands) of the visible parts of the four RSV P protomers in the cryo-EM structure of the RSV L—P complex (PDB 6PZK, [41]).

**Figure 2 biomolecules-11-01225-f002:**
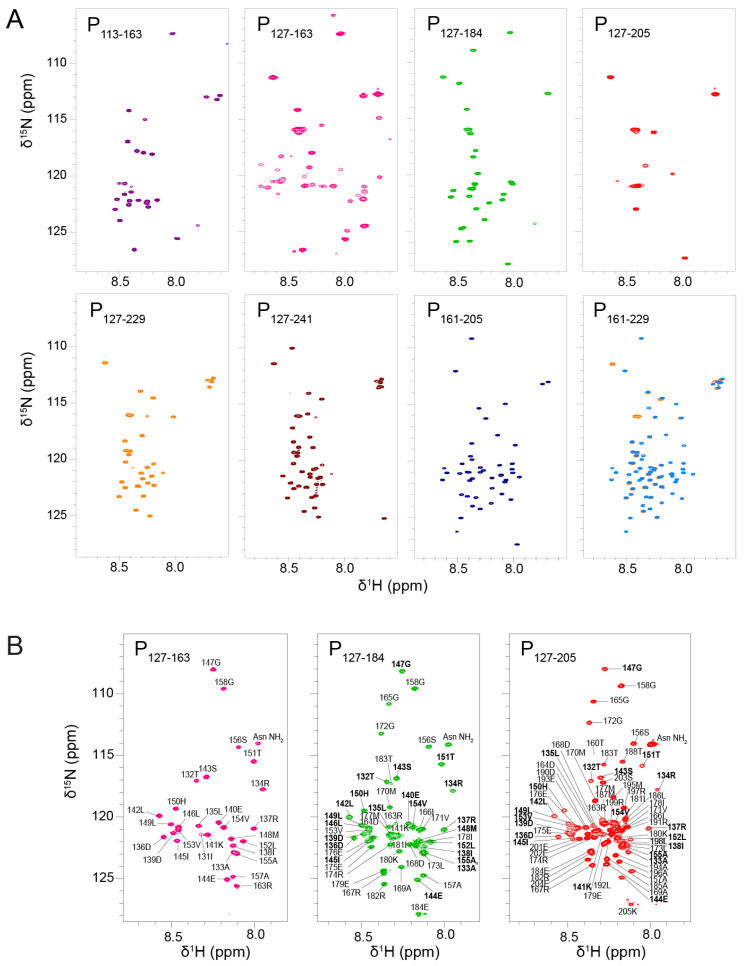
NMR fingerprints of RSV phosphoprotein fragments. (**A**) 2D ^1^H-^15^N HSQC spectra of eight ^15^N-labeled RSV P fragments were recorded at low temperature (288 K) in 20 mM phosphate pH 6.5, 100 mM NaCl buffer and at 150–300 µM concentrations. These fragments either contain the oligomerization domain (P_OD_) with an N-terminal extension (P_113–163_), only P_OD_ (P_127–163_), P_OD_ with C-terminal extensions of different lengths (P_127–184_, P_127–205_, P_127–229_ and P_127–241_) or only C-terminal extensions (P_161–205_ and P_161–229_). NMR data were measured at 600 MHz ^1^H Larmor frequency for all fragments, except P_113-163_ that was measured at 700 MHz. Each fragment was assigned a color: P_113–163_ (purple), P_127–163_ (pink), P_127–184_ (green), P_127–205_ (red), P_127–229_ (orange), P_127–241_ (maroon), P_161–205_ (marine), and P_161–229_ (medium blue). The spectrum of P_127–229_ (orange) was superimposed onto that of P_161–229_ (blue). Spectra were plotted with comparable contour levels. (**B**) 2D ^1^H-^15^N HSQC spectra of three ^15^N-labeled RSV P fragments were measured at high temperature (313 K) on a 600 MHz NMR spectrometer. Amide signals are annotated with residue number and amino acid type. P_OD_ residues are highlighted in bold letters.

**Figure 3 biomolecules-11-01225-f003:**
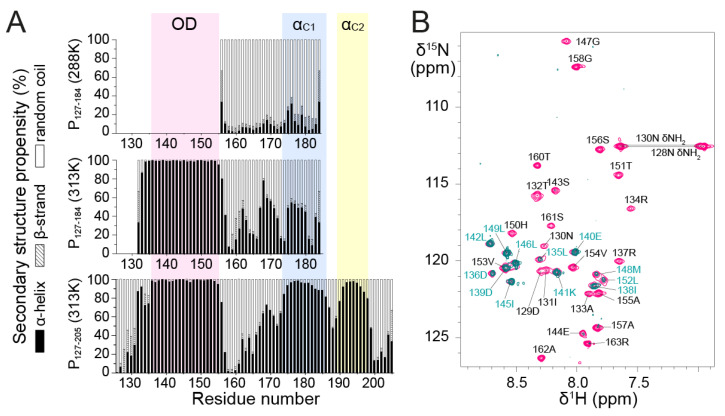
NMR data and secondary structure propensities of RSV phosphoprotein fragments containing P_OD_ and C-terminal extensions of variable lengths. (**A**) Secondary structure propensities for α-helix, β-strand and random coil were determined from backbone chemical shift by using the TalosN server [47]. ^13^Cα, ^13^CO, ^13^Cβ and ^1^Hα values were used as an input for P_127–184_ and P_127–205_ at 313 K. Only ^13^Cα values were available for P_127–184_ at 288 K. (**B**) Amide ^1^H/^2^H exchange experiment by NMR. 2D ^1^H-^15^N HSQC spectra of ^15^N-labeled P_127–163_ before (pink) and after exchange into 100% ^2^H_2_O buffer (teal) were overlaid. Measurements were carried out at a temperature of 298 K on an 800 MHz NMR spectrometer. Assignments for amides that are protected from water exchange are indicated in cyan.

**Figure 4 biomolecules-11-01225-f004:**
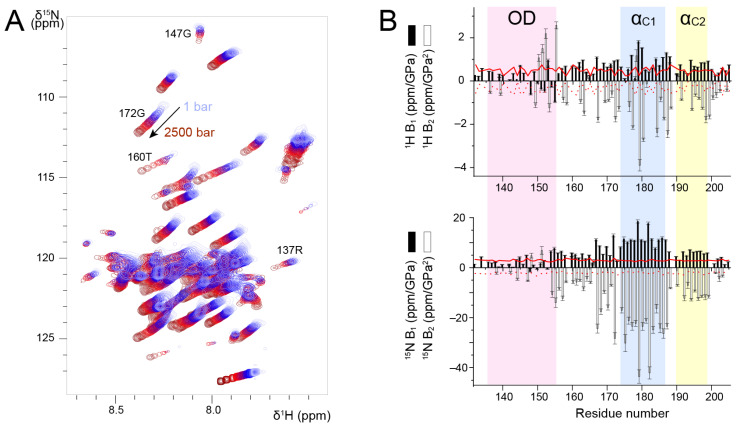
Discrimination between stably associated and isolated helices in RSV P_127–205_ by variable pressure NMR experiments. (**A**) Overlay of ^1^H-^15^N HSQC spectra of P_127–205_ acquired by increasing pressure from 1 to 2500 bar. NMR data were recorded at 600 MHz ^1^H Larmor frequency. The temperature was 313 K. (**B**) First and second order pressure coefficients, B_1_ and B_2_, were determined by fitting ^1^H and ^15^N chemical shift variations as a function of pressure to a second-order polynomial according to δ(P) − δ(P_0_) = B_1_ × (P − P_0_) + B_2_ × (P − P_0_)^2^, and plotted with bar diagrams. Reference values for random coil conformation (taken from [60]) were plotted as red lines (B_1_, solid line) and (B_2_, dotted line).

**Figure 5 biomolecules-11-01225-f005:**
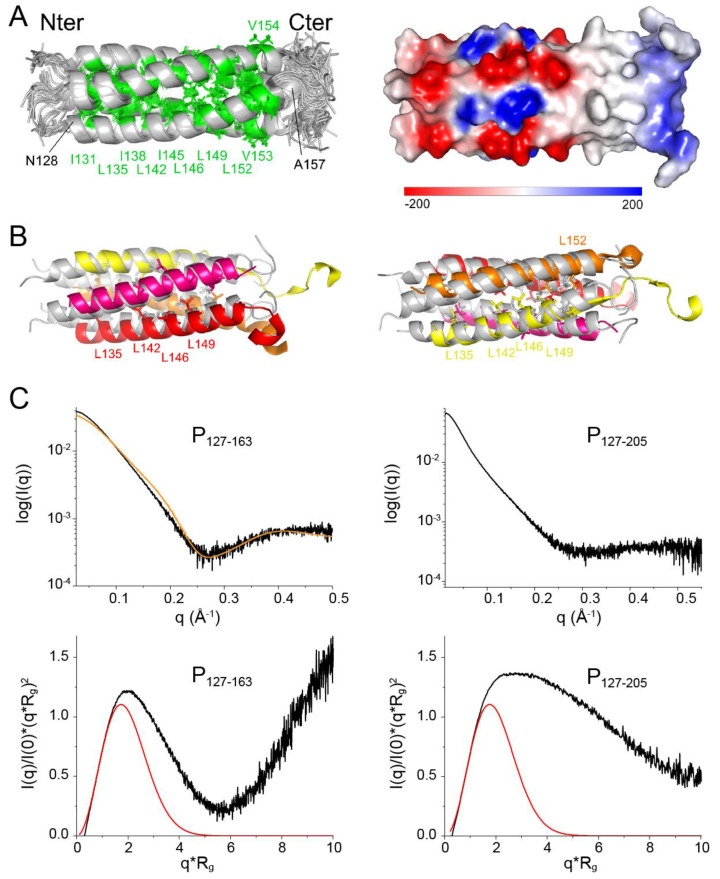
Solution structure of the oligomerization domain of RSV P and dynamics of its C-terminal extensions. (**A**) NMR ensemble structure of RSV P_127–163_. 20 conformer structures calculated with the ARIA software [48] were structurally aligned. Backbones are represented in cartoon and the three hydrophobic residues Ile, Leu and Val in green sticks. The electrostatic surface potential was calculated using the Delphi software [62]. The scale for the ensemble of 20 structures ranges from −200 k_B_T (red) to +200 k_B_T (blue). (**B**) Superimposition of the NMR structure of RSV P_127–163_ (conformer 1, in grey) onto the cryo-EM structure of the same sequence in the P—L complex (PDB 6pzk). Each protomer has a different color: yellow, orange, magenta and red. Leucine side chains are shown in sticks. (**C**) SAXS data were measured for P_127–163_ and P_127–205_. The scattering intensity curves (log(I) = f(q)), where q is the scattering vector amplitude) are shown. The scattering intensity curve corresponding to the NMR structure of P_127–163_ (orange line) was calculated with the FoXs server [55] and superimposed onto the experimental data. Normalized Kratky plots ((q*R_g_)^2^*I(q)/I(0) = f(q*R_g_)) were calculated using the R_g_ and I(0) values obtained with the PRIMUS software [54]: 21.2 Å and 0.0447 for P_127–163_; 36.8 Å and 0.0749 for P_127–205_. The theoretical normalized Kratky plot for a globular protein is plotted with a red line.

**Figure 6 biomolecules-11-01225-f006:**
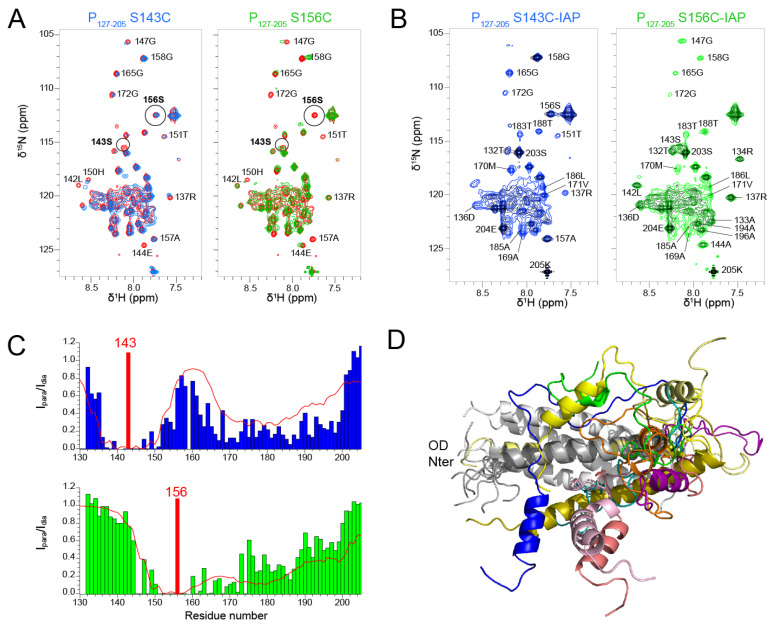
Spatial sampling of the C-terminal P_Cα_ region proximal to the oligomerization domain of RSV P by Paramagnetic Relaxation Enhancement experiments. (**A**) Comparison between ^1^H-^15^N HSQC spectra of WT P_127–205_ (red) and P_127–205_ mutants S143C (blue) and S156C (green). NMR spectra were acquired at 600 MHz ^1^H frequency and a temperature of 313 K. Assignments of several isolated signals are indicated. (**B**) ^1^H-^15^N HSQC spectra of IAP-labeled P_127–205_ S143C and S156C mutants in paramagnetic (black) and diamagnetic states (S143C in blue, S156C in green). (**C**) Paramagnetic relaxation enhancements (PRE), determined as intensity ratios I_para_/I_dia_ for each residue, are plotted as bar diagrams. The position of the IAP spin labels is indicated by a red bar. Theoretical PREs, plotted as a red line, were calculated with the Flexible-meccano software [50] from an ensemble of 1000 P_127–205_ structures, using a contact distance of 7 Å between P_OD_ residues 143–156 and P_Cα_ residues 160–199. (**D**) 10 randomly chosen P_127–205_ conformers were aligned onto the P_OD_ structure to exemplify the transient structure of P_Cα_ and the transient contacts between the P_Cα_ helices and P_OD_.

**Figure 7 biomolecules-11-01225-f007:**
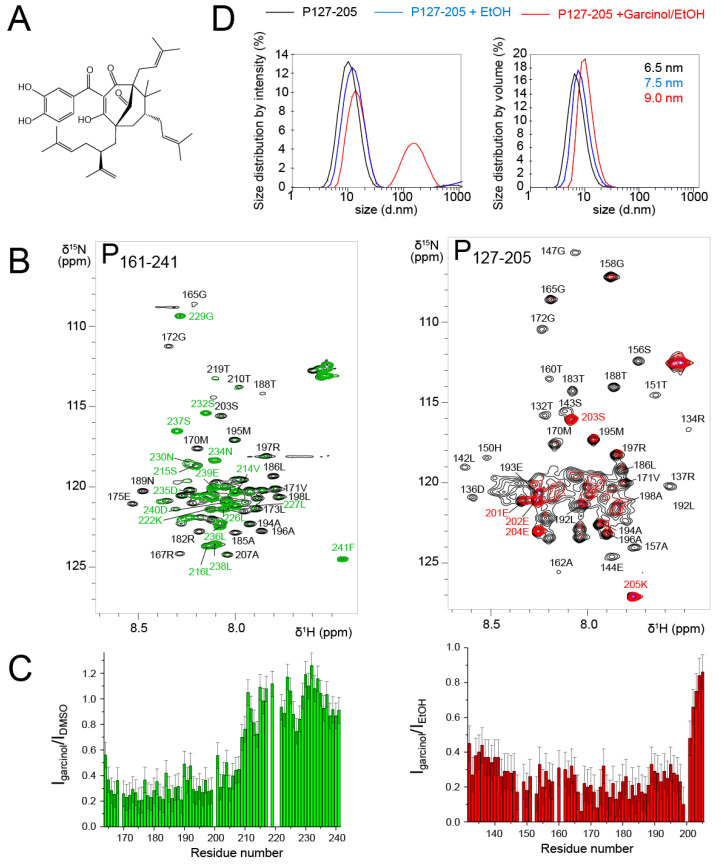
Effect of garcinol on the dynamics of RSV P_127–205_ probed by NMR and DLS. (**A**) Chemical formula of garcinol. (**B**) Comparison between ^1^H-^15^N HSQC spectra of P_161–241_ (T = 293 K, 150 µM) before (black) and after addition of 3 molar equivalents of garcinol in DMSO (green) and of P_127–205_ (T = 313 K, 300 µM) before (black) and after 1 molar equivalent of garcinol in ethanol (red). (**C**) Intensity ratios, represented as bar diagrams, were determined from the intensities of ^1^H-^15^N HSQC spectra of P_127–205_ and P_161–241_ in the presence of garcinol versus solvent. Error bars correspond to the values of the RMSD determined in the regions with line broadening. (**D**) Size distribution in P_127–205_ samples by dynamic light scattering.

## Data Availability

Data is contained within the article.

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
