# Peer review of "A Structural and Dynamic Analysis of the Partially Disordered Polymerase-Binding Domain in RSV Phosphoprotein"

_biomolecules, 2021, doi:10.3390/biom11081225_

Round 1

Reviewer 1 Report

The text could be more concise, it is very voluminous and complicated.

Otherwise, the paper is sound and doesn't need modifications.

Typo: in line 664, instead of FigureFigure6 just Figure 6.

The paper wants to shed light on the structural properties of the POD-PCa region of phosphoprotein P in solution by NMR and other biophysical methods. It is an actual question and the answer is interesting. The topic has already been dealt with but important details remained unanswered which are now addressed. They investigated in more detail the structure and dynamics of the consecutive POD and PCa domains in unbound respiratory 16 syncytial virus phosphoprotein as a novelty. The paper is well written but more concise texting would be advantageous. The text is clear and easy to read. The conclusions are consistent with the evidence and arguments presented and they address the main questions posed. 

Author Response

We thank the reviewer for the thorough reading and the positive comments. We corrected the typo. We agree that the paper may have been written more concisely, but since the reviewer also stated that the text was clear, we chose not to shorten it.

Reviewer 2 Report

The paper present efforts on analyzing the role of Pcα domain of RSV phosphoprotein P. The Pcα domain could not be fully characterized by NMR, so the authors using small phosphoprotein fragments centered on or adjacent to Pod domain to recover the NMR signal. Combining with PRE, SAXS measurement and reported Cryo-EM data, the authors proposed the possible structure property and role of Pcα.  Then the effect of small molecule garcinol were tested, which revealed that garcinol could stabilize the Pcα helices and/or promotes interprotomer or intermolecular interactions between these helices. The experiment design and result are rational and clear. I think this paper is qualified to be published with minor revisions.

Detailed comments:

  1. Please show the superimposition of P127-163 NMR structure with reported cryo-EM structure.
  2. It seems that garcinol has significant effect on the whole structure of Pcα, why?  Whether it is possible that garcinol bind to Pcα with a molar ratio > 1 ? 

Author Response

We thank the reviewer for the thorough reading and the positive comments. According to her/his advice, we added the superimposition of the P127-163 NMR structure with the same part in the cryo-EM structure of the P-L complex to Figure 5.

To answer the question about the effect of garcinol on the structure of the PCα domain, we interpret our data as an effect of intra-helix association. In this process the helical structure of PCα  is likely stabilized. We could not determine a precise binding site, and the stoichiometry could be >1. To clarify this point, we completed the sentence starting at line 606: "This would imply that garcinol stabilizes the P helices and/or promotes interprotomer or intermolecular interactions between these helices, leading to the observed effect." and added following sentence at line 608: "Our experiments did not allow to determine the binding site of garcinol nor the stoichiometry of the putative garcinol-P complex."

Reviewer 3 Report

In this manuscript, Cardone et al observed the novel finding that the structural property of Pca along with POD, constitutes the basis for the peculiar binding mode of RSV phosphoprotein to L. The authors have performed a structural and dynamic analysis of the Pca-POD domain in RSV phosphoprotein thereby demonstrating that each of the four protomers binds to L in a different way. Overall, the data are supportive of the conclusions reached and the information has novelty and significance for the journal's readership. It can be accepted as it is.

Author Response

We thank the reviewer for the thorough reading and the very positive comments.